# Quantifying the differential functional behavior between the medial and lateral meniscus after posterior meniscus root tears

**Brian E. Walczak** *, **Kyle Miller, Michael A. Behun, Lisa Sienkiewicz, Heather Hartwig Stokes, Ron McCabe, Geoffrey S. Baer**

Department of Orthopedics & Rehabilitation, Advancement of Translational Orthopedics and Medical Sciences (ATOMS) Laboratory, Wisconsin Institute of Medical Research (WIMR), University of Wisconsin-Madison, Madison, WI, United States of America

* walczak@ortho.wisc.edu

**Data Availability Statement:** Data are published through DRYAD and accessible at: https://doi.org/10.5061/dryad.1ns1rn8v9.

## Abstract

Meniscus tears of the knee are among the most common orthopedic knee injury. Specifically, tears of the posterior root can result in abnormal meniscal extrusion leading to decreased function and progressive osteoarthritis. Despite contemporary surgical treatments of posterior meniscus root tears, there is a low rate of healing and an incidence of residual meniscus extrusion approaching 30%, illustrating an inability to recapitulate native meniscus function. Here, we characterized the differential functional behavior of the medial and lateral meniscus during axial compression load and dynamic knee motion using a cadaveric model. We hypothesized essential differences in extrusion between the medial and lateral meniscus in response to axial compression and knee range of motion. We found no differences in the amount of meniscus extrusion between the medial and lateral meniscus with a competent posterior root (0.338mm vs. 0.235mm; p-value = 0.181). However, posterior root detachment resulted in a consistently increased meniscus extrusion for the medial meniscus compared to the lateral meniscus (2.233mm vs. 0.4705mm; p-value < 0.0001). Moreover, detachment of the posterior root of the medial meniscus resulted in an increase in extrusion at all angles of knee flexion and was most pronounced (4.00mm ± 1.26mm) at 30-degrees of knee flexion. In contrast, the maximum mean extrusion of the lateral meniscus was 1.65mm ± 0.97mm, occurring in full extension. Furthermore, only the medial meniscus extruded during dynamic knee flexion after posterior root detachment. Given the differential functional behaviors between the medial and lateral meniscus, these findings suggest that posterior root repair requires reducing overall meniscus extrusion and recapitulating the native functional responses specific to each meniscus.

## Introduction

The menisci are fibrocartilage structures that distribute load across the knee joint's articular cartilage during activity [1]. Tears of the meniscus are among the most common orthopedic knee

**Funding:** B.E.W. was supported by National Center for Advancing Translational Sciences under award numbers: UL1 TR000427, TL1 TR000429, and TL1 TR002375, National Institute on Aging under award number: T32 AG000213, and the Department of Orthopedic Surgery of the University of Wisconsin-Madison's Freedom of Movement Award. The funders had no role in study design, data collection and analysis, decision to publish, or preparation of the manuscript.

**Competing interests:** B.E.W. is a consultant for AlloSource and G.S.B. is a consultant for Conmed. This does not alter our adherence to PLOS ONE policies on sharing data and materials. The commercial organizations did not play a role in the study design, data collection and analysis, decision to publish, preparation of the manuscript, and provided no financial support or compensation.

injuries, with an annual incidence of approximately 61 per 100,000 persons [2]. Posterior root tears (PRTs) are a specific type of meniscus tear resulting in the detachment of the posterior root ligament resulting in extrusion of the meniscus from between the femur and tibia [3]. Previous studies demonstrate biomechanical alterations of the knee after PRT resulting from increased meniscus extrusion leading to subsequent osteoarthritis (OA) [4–6]. Posterior root repair offers a promise of restoring normal contact forces across the articular cartilage [5] and improving clinical outcomes in the short term [7]. However, a critical challenge to be resolved is the recapitulation of native meniscus function and prevention of knee OA [8–10]. There have been considerable efforts to minimize meniscus extrusion by improving surgical techniques and fixation strength of the suture repair; however, there remains a high rate of postoperative meniscus extrusion [4, 11–14].

Furthermore, recent studies suggest that there are essential differences in PRTs between the medial and lateral meniscus. For instance, medial meniscus PRTs are associated with degenerative changes and advanced age [15], while lateral meniscus PRTs are most commonly associated knee trauma, such as anterior cruciate ligament ruptures [16]. In a recent clinical study comparing the medial and lateral meniscus PRTs, Koo et al. found a higher rate and increased amount of meniscus extrusion of the medial meniscus compared to the lateral meniscus on static magnetic resonance imaging (MRI) after PRTs [1].

To demonstrate the effects of knee activity on meniscus motion, here we compared meniscus extrusion of the medial and lateral meniscus during axial compression and dynamic motion before and after detachment of the posterior root. The objective of this study was to understand essential differences in response to PRTs between the medial and lateral meniscus.

## Materials and methods

### Cadaveric dissection and posterior root detachment

The University of Wisconsin-Madison Institutional Review Board approved the study under 2016–1316, and as a cadaveric study with anonymous data analysis, written consent was not required. Human knees were obtained from the Anatomic Gifts Registry. Knee specimens were cleared of soft tissue leaving the capsule intact as previously described by Allaire et al [8]. The femur length was reduced to 6-inches and the tibia length to 4-inches to accommodate testing. Holes were drilled through the distal end of the bones for pin placement to help fixate the potting to the bone. Shafts were pinned with 3.2-mm steel pins and potted in tubular molds using auto body filler. The molds were 44-mm in diameter for the femur and 56-mm for the tibia to fit in custom fabricated grips. The condition of a torn posterior meniscus root was simulated by transection of the meniscus within 1-cm of the root attachment using an arthroscopic meniscus knife as described by LaPrade et al [17]. For the lateral meniscus, the meniscal femoral ligaments were left intact.

### Equipment and setup

Custom fabricated grips were attached to an MTS 858 Bionics servo-hydraulic test system (Fig 1). The lower grip was fabricated using an axle system to release the knee rotational (medial/lateral) degree of freedom. The axle system was then mounted on two crossed roller slides (Deltron NBT-6160A) in an X-Y arrangement to allow degree of freedom release of the medial/lateral and anterior/posterior planes (Fig 1A). Data acquisition included the specimen load, actuator displacement, and meniscus displacement, using a Lord Microstrain model M-DVRT-6 transducer (M-DVRT-6, 6.0mm stroke micro-miniature DVRT linear displacement Transducer, and DEMOD-DC, Miniature DC input, DC output signal conditioner, LORD Corporation, 459 Hurricane Lane, Suite 102, Williston, Vermont 05495) (Fig 1B). Data were acquired using a Data Translation 16-bit digital to analog converter. The force generated

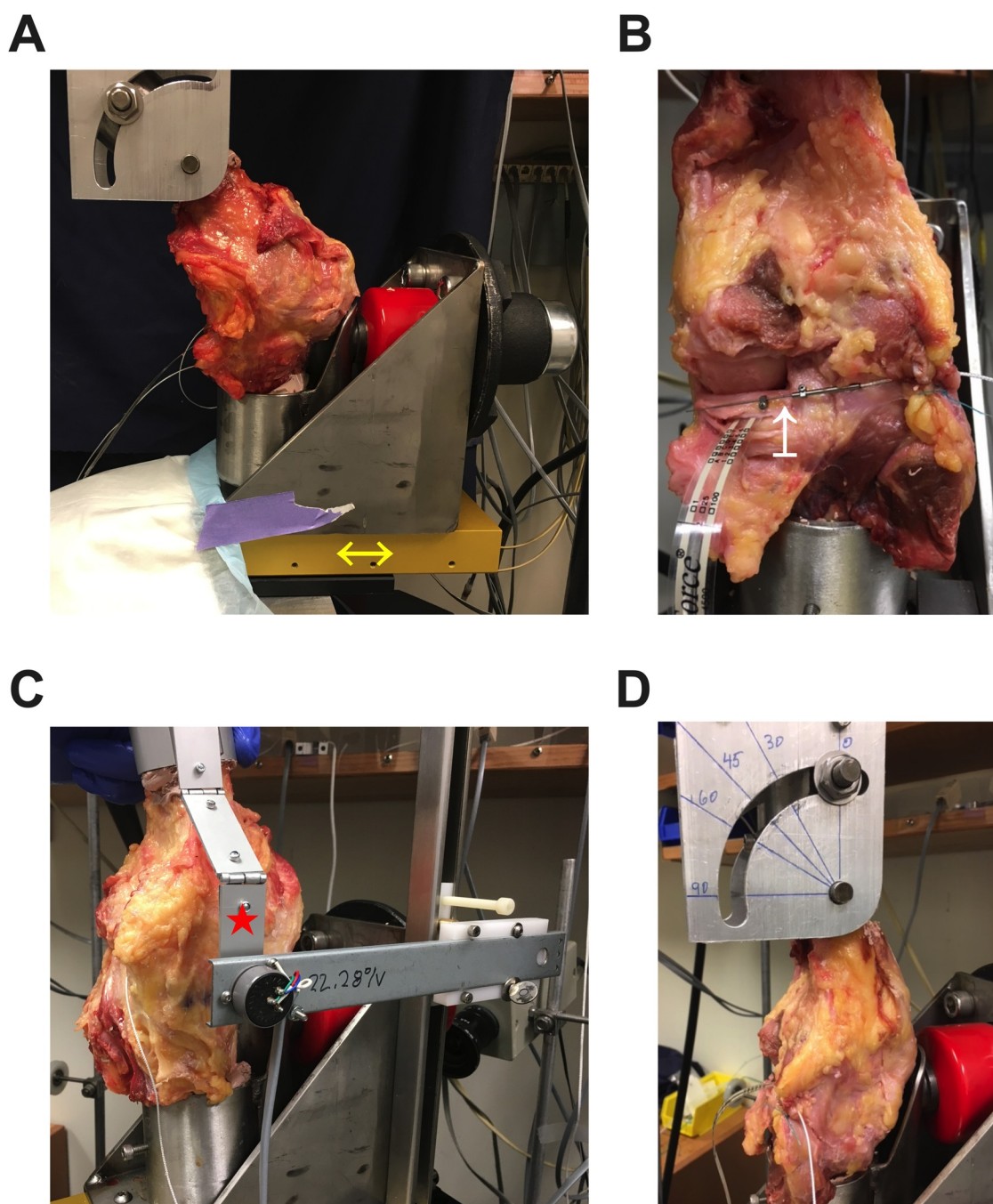

**Fig 1. Photographs of MTS testing setup.** (A) Custom base (yellow bidirectional arrow) allowed for three degrees of freedom, and femoral potting (top) allowed for control of knee flexion. (B) Photograph of cadaveric testing. A displacement sensor (white arrow) was placed at the posterior meniscus root attachment 1-cm from the insertional footprint and in the tibia's periosteum at the knee's midline (capsule reflected only for demonstration purposes). (C) Photograph demonstrating testing setup during knee range of motion. An electronic goniometer arm (red star) captured real-time (60 Hz) knee angle during range of motion testing. (D) Photograph of the calibrated femoral potting (top) used for axial compression testing.

during each testing protocol was recorded in real-time. The upper grip was fabricated to allow positioning the knee in the range of zero to one hundred degrees of flexion for mechanical testing during a dynamic range of motion (Fig 1C).

## Testing protocols

Protocols simulated clinical knee activities during weight-bearing, including axial compression loading and passive non-weight-bearing dynamic range of motion. The amount of load delivered across the knee simulated that of a 75-kg person [18–20]. Specimens were tested in the following sequence: posterior root intact and posterior root detached. *Axial compression load*: Testing was performed in load control in the axial direction. A ramp to a constant load was completed to simulate a load from 25-N preload to full weight-bearing (1800-N) at 0.1-second ramp to set load, with a manual return to preload in full knee extension (0-degrees of knee flexion). A sinusoidal cyclic load to simulate a walking gait at 20-cycle compressive ½ cycle sine wave, 25-N to set load with 1-sec cycle (0.3-second down, 0.3-second up, 0.4-second pause to stimulate swing phase of gait). This test was performed across 0-, 30-, 45-, 60- and 90-degrees of knee flexion. *Dynamic non-weight-bearing range of motion*: Each knee was taken through a range of motion from 0- to 100-degrees of flexion. Meniscus extrusion was linked to range of motion data detected with a calibrated custom electronic goniometer in real-time.

## Statistical analysis

Means and standard deviations or standard error of the means were calculated. Between group comparisons were performed using a Student's t-test for two groups and analysis of variance (ANOVA) for more than two groups. Pairwise comparisons and post-hoc adjustments were performed if the ANOVA model was statistically significant. Linear regression was used to model the relationship between knee flexion and meniscus extrusion. A p-value < 0.05 was considered statistically significant. Calculations were performed using GraphPad Prism 9.0.1 for Mac (San Diego, CA, USA).

## Priori power analysis

Based upon our preliminary data demonstrating a variance of 3.0-mm between groups and a variance of 1.27-mm within groups, a total of 3.27 meniscus roots per group would need to be tested to achieve 80% with a balanced one-way ANOVA with a 5% significance level.

## Results and discussion

This study used a total of eleven cadaveric specimens (six for testing the medial meniscus and five for testing the lateral meniscus). There were six males and five females, seven right knees and four left knees, with a mean age of 69.7 years (range: 36–84 years) (S1 Table). The overall amount of extrusion with a competent posterior meniscus root did not differ between the medial and lateral meniscus. However, overall medial meniscus extrusion was increased compared to the lateral meniscus in response to detachment of the posterior root (Table 1). Data is available at doi:10.5061/dryad.1ns1rn8v9 [21].

   With a competent posterior root, a constant axial compressive load across the knee in full extension result in a mean extrusion of 0.436mm (± 0.459mm) for the medial meniscus and 0.270mm (± 0.250mm) for the lateral meniscus (p-value = 0.893). Detachment of the posterior meniscus root resulted in increased meniscus extrusion for both the medial (Fig 2A) and lateral meniscus (Fig 2B). Extrusion of the medial meniscus was increased compared to the lateral meniscus after posterior root detachment (Fig 2C).

   To test the hypothesis that meniscus extrusion is a function of knee flexion, knees were subjected to cyclic axial compression loading at knee flexion angles at 0-, 30-, 45-, 60-, and 90-degrees. Overall, the mean extrusion of an intact medial meniscus was seen at 45-degrees of knee flexion (0.7235mm ± 0.298mm). However, the mean extrusion of an intact medial

**Table 1. Summary data for medial and lateral posterior root extrusion[a].**

| Condition | Medial Meniscus | Lateral Meniscus | p-value[b] |
|---|---|---|---|
| Intact | 0.338 ± 0.296 | 0.235 ± 0.242 | 0.181 |
| Torn | 2.233 ± 1.270 | 0.4705 ± 0.566 | **0.001** |

[a] Data represents mean ± standard deviation. n = 11 aggregated across all testing conditions.

[b] Student's t-test comparing medial meniscus and lateral meniscus displacement. Bold indicates statistical significance.

meniscus was not dependent on the degree of knee flexion (p-value = 0.304). Conversely, detachment of the posterior root resulted in an increase in extrusion of the medial meniscus across all tested degrees of knee flexion (Fig 3A). The greatest extrusion was recorded at 30-degrees of knee flexion (4.000 ± 01.259mm; p-value = 0.015). Loading the knee in axial compression at various degrees of knee flexion resulted in no difference in extrusion of an intact lateral meniscus (p-value = 0.787). Knee flexion of 45-degrees resulted in the greatest extrusion of an intact lateral meniscus (0.235mm ± 0.007mm). In agreement with the previous findings of extrusion of the lateral meniscus after posterior root detachment under constant axial compression loading in full knee extension, cyclic loading resulted in an increase in lateral meniscus extrusion after posterior root detachment at 0-degrees of knee flexion. However, none of the other knee flexion categories resulted in increased extrusion of a detached posterior lateral meniscus root compared to an intact root (Fig 3B). Although there was no difference in extrusion of an intact medial meniscus posterior root compared to a lateral meniscus, detachment of the posterior root resulted in increased extrusion of the medial meniscus compared to the lateral meniscus at all degrees of knee flexion (Fig 3C). Thirty-degrees of knee

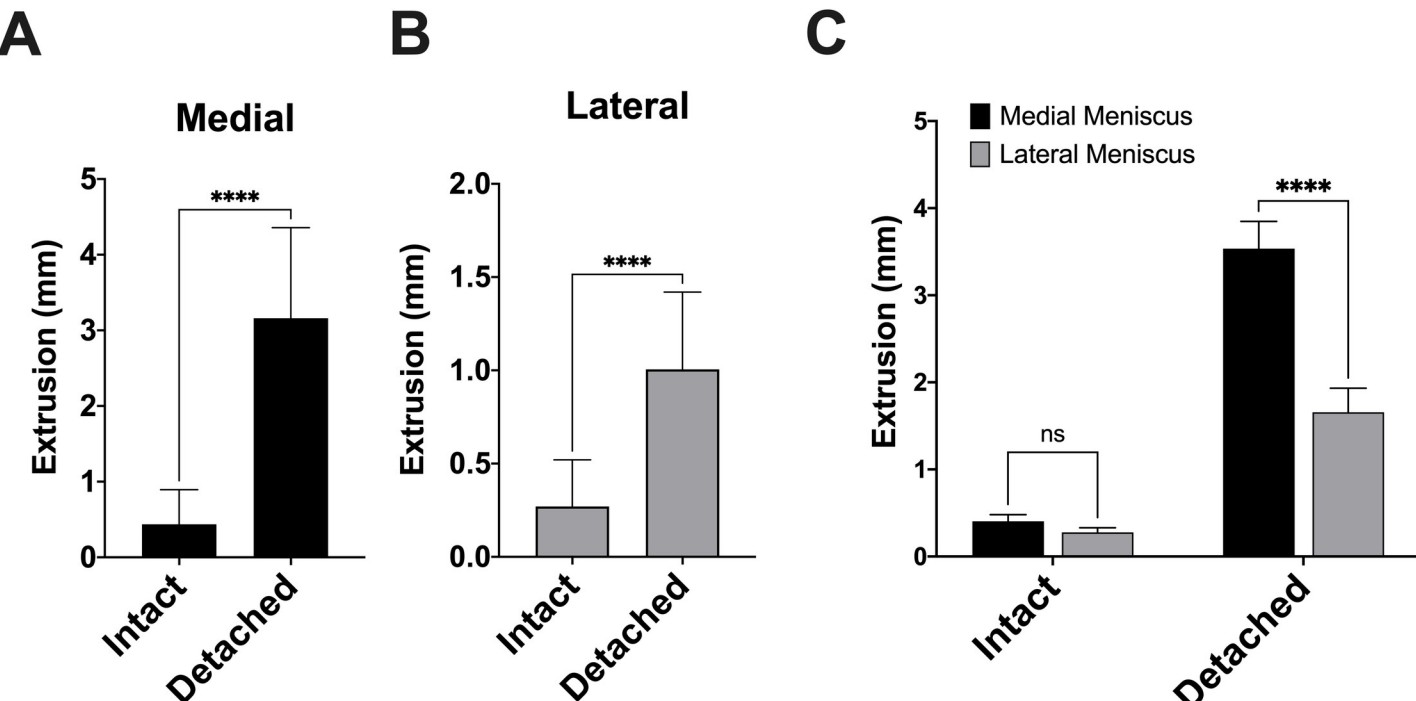

**Fig 2. Comparison of meniscus extrusion between the medial and lateral meniscus under constant axial compression load.** (A) Meniscus extrusion comparing intact versus detached condition for the medial meniscus. (B) Meniscus extrusion comparing intact versus detached condition for the lateral meniscus. (C) Comparison of meniscus extrusion between the medial and lateral meniscus. n = 5 cadavers aggregated in triplicate. ns = not statistically significant; **** p-value < 0.0001.

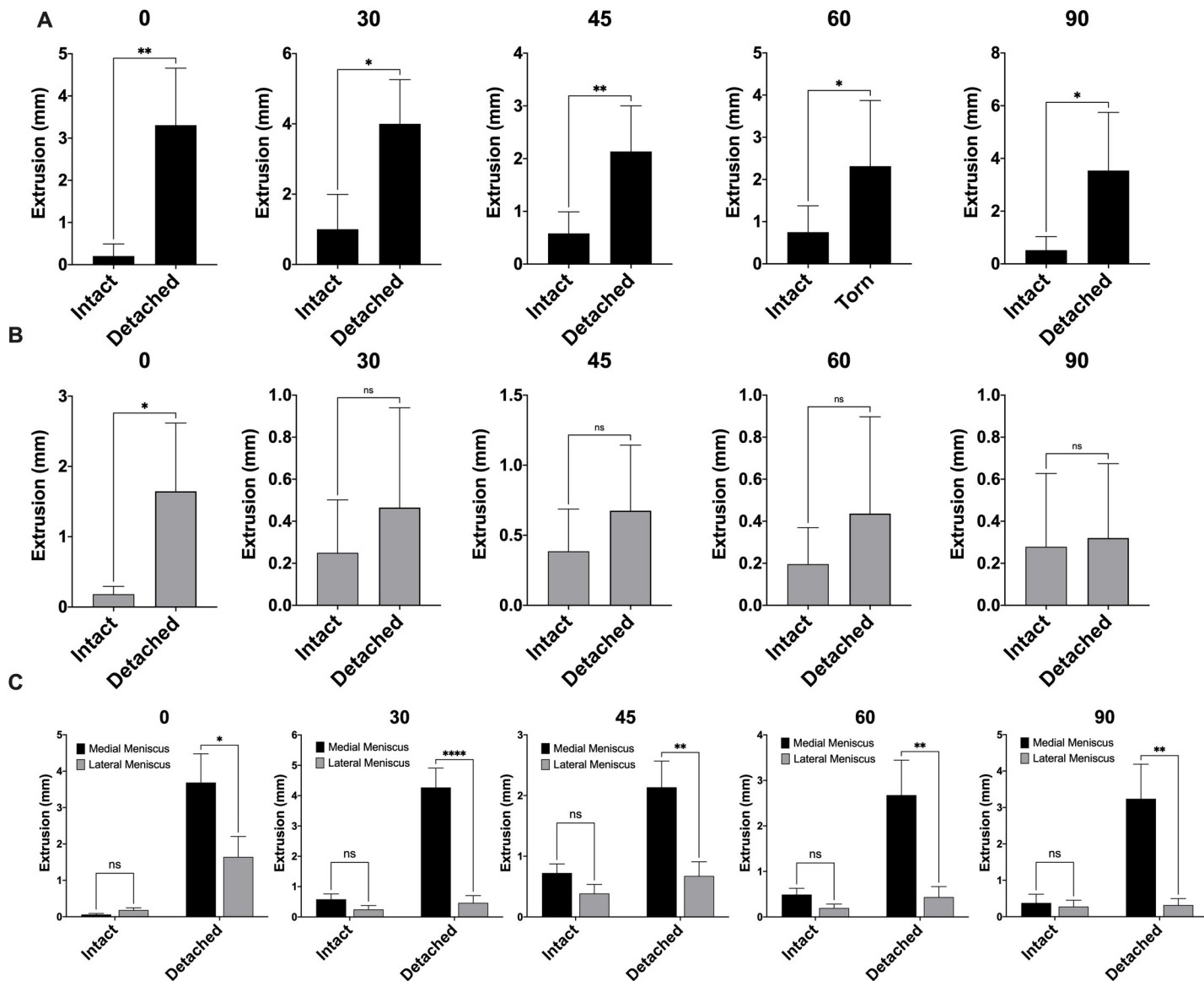

**Fig 3. The effects of knee flexion on meniscus extrusion during axial compression loading.** (A) Comparison of meniscus extrusion between intact and detached posterior meniscus root of the medial meniscus. (B) Comparison of meniscus extrusion between intact and detached posterior meniscus root of the lateral meniscus. (C) Comparison of meniscus extrusion between medial and lateral meniscus. n = 5. Titles represent the angle of knee flexion in degrees. ns = not statistically significant; * p-value < 0.05; ** P-value < 0.01; **** p-value < 0.0001.

flexion resulted in the greatest difference between the medial and lateral meniscus extrusion of a torn posterior root (3.805mm ± 0.0.505mm; p-value < 0.0001).

To test the hypothesis that cyclic range of knee motion without axial compression load results in an increase in meniscus extrusion, cadaveric knees were subjected to dynamic range of motion from full extension to maximum allowable flexion capturing meniscus extrusion and range of motion with an electric goniometer in real-time. There was a direct linear relationship between increasing knee flexion and meniscus extrusion with an intact posterior root for both the medial and lateral meniscus (Fig 4A). However, detachment of the posterior root resulted in a direct nonlinear relationship between increasing knee flexion and meniscus extrusion for the medial meniscus with a maximum of 4.59mm of extrusion occurring at

19.4-degrees of knee flexion and 2.69mm occurring at 63.8-degrees of knee flexion for the lateral meniscus (Fig 4B). In comparison, detachment of the lateral meniscus posterior root resulted in increased variability in meniscus extrusion, but a direct linear relationship between knee flexion and meniscus extrusion was preserved (Fig 4B). Means and standard deviations were calculated and compared between the medial and lateral meniscus for each 10-degrees of knee flexion (Fig 4C). The mean extrusion of an intact posterior medial root was greater than an intact posterior lateral root for knee flexion greater than 40-degrees, although this did not reach the threshold of statistical significance. However, detachment of the posterior root resulted in increased extrusion of the medial meniscus compared to the lateral meniscus for knee flexion greater than 10-degrees.

Tears of the posterior meniscus roots account for up to 21% of all meniscus tears [3]. PRTs of the medial meniscus treated with partial meniscectomy have a conversion rate to total knee replacement of up to 35% [22], and PRTs of the lateral meniscus are associated with anterior crucial ligament tears [1, 16]. The stark association between PRT and clinically progressive disease is purportedly due to the functional consequences, namely meniscus extrusion and the resultant abnormal contact stresses across the knee joint [8, 23]. For example, one study found that a PRT of the medial meniscus caused a 25% increase in peak contact pressure compared to an intact meniscus [8]. A more recent study used a finite element analysis to demonstrate that a PRT of the lateral meniscus caused maximum contact pressure and contact stress values approaching that found after total lateral meniscectomy [23].

Although medial meniscus PRTs treated without repair are associated with progressive joint degeneration, results of surgical repair are not uniformly superior. For example, after medial meniscus repair, MRI evaluation has demonstrated continued meniscus extrusion [24] and low healing rates [25]. Similarly, the paucity of research regarding the clinical implications of PRTs of the lateral meniscus demonstrates lax healing of up to 21% [26], and a recent systematic review was unable to conclude whether root repair restored the functional hoop stress characteristics required of the meniscus to distribute compression forces across the knee evenly [27]. Furthermore, a recent study found characteristic clinical differences between the medial and lateral meniscus PRTs, including the association of trauma in lateral meniscus PRTs, and older age and higher grade of OA changes of the knee in medial meniscus PRTs [1]. Although surgical repair strategies can repair the posterior root to the tibia, the recapitulation of meniscus function remains a significant challenge in clinical orthopedics. Here we examined the behavior of the meniscus in response to both axial compression load and dynamic range of motion of the knee.

Our model successfully captured the amount of meniscus extrusion under axial compression and knee motion, as evidenced by the increased extrusion amount after detachment of the posterior root in both medial and lateral menisci. Interestingly, the findings of increased meniscus extrusion after detachment of the posterior root in both the medial and lateral meniscus suggest the loss of structural integrity and the possibility of compromised resistance to hoop stress after PRTs of both medial and lateral meniscus. Indeed, the detection of meniscus extrusion of MRI has been reported to be evident after PRT clinically [1, 4, 28]. Moreover, the degree of extrusion is associated with the progression of articular cartilage damage for the medial meniscus [10].

Here, we found no difference in meniscus extrusion with an intact posterior root between the medial and lateral meniscus under a constant axial compression load across a fully extended knee. However, detachment of the posterior root resulted in a significant increase in both the medial and lateral meniscus extrusion compared to an intact posterior root. Interestingly, extrusion of the medial meniscus was increased compared to the lateral meniscus after posterior root detachment. These results are consistent with Koo et al., who reviewed MRIs of

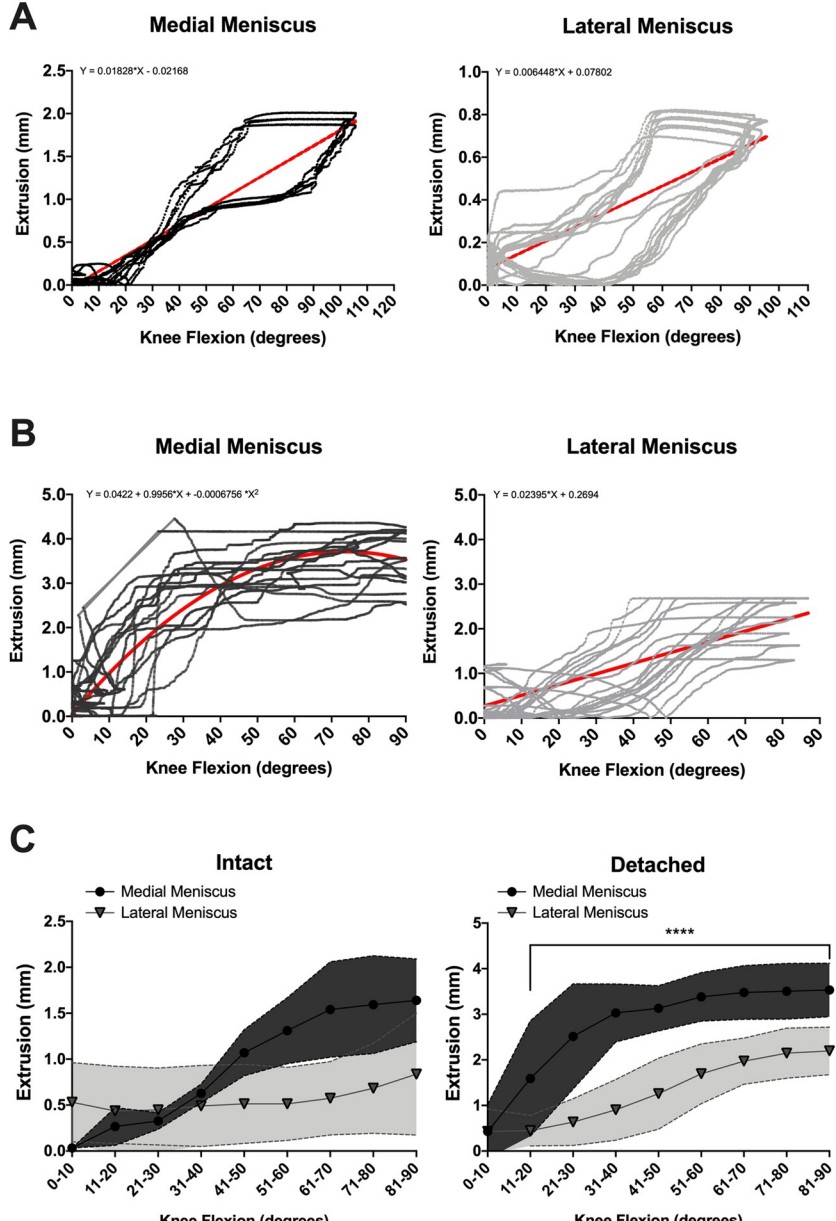

**Fig 4. The effects of cyclic range of motion without axial compression on meniscus extrusion.** (A) Scatterplot of the relationship of knee range of motion and meniscus extrusion for the medial and lateral meniscus with an intact posterior root. (B) Scatterplot of the relationship of knee range of motion and meniscus extrusion for the medial and lateral meniscus with the posterior root detached. (C) Comparison of meniscus extrusion between the medial and lateral meniscus. n = 5. Top left: Regression equation represented by the red line. **** p-value < 0.0001.

patients with PRTs and found mean meniscal extrusion of 4.2mm for the medial meniscus and 0.9mm for the lateral meniscus [1].

The effects of knee flexion on meniscus extrusion are not well described. Previous clinical research has examined MRI findings after PRTs [1, 4, 28–30], and biomechanical protocols aim to test suture repair constructs in a fixed knee position [5, 12, 31–35]. Again, we found small amounts of extrusion when the posterior root was intact with no difference between the medial and lateral meniscus regardless of knee flexion angle. Similarly, there were differences

in the medial and lateral meniscus behavior after detachment of the posterior root. The medial meniscus extrusion increased after posterior root detachment at every knee flexion angle. The greatest amount of extrusion was found at 30 degrees of knee flexion (mean 4.00mm ± 1.26mm). However, the lateral meniscus extruded the greatest amount after posterior root detachment in full extension (mean 1.645mm ± 0.971mm). Moreover, we found that lateral meniscus extrusion did not increase significantly after posterior root detachment when subjected to axial compression load at greater angles of knee flexion. Compared to the lateral meniscus, the medial meniscus demonstrated increased extrusion at all knee flexion angles (mean difference 2.49mm ± 0.90mm). These findings suggest that compared to the lateral meniscus, extrusion of the medial meniscus may increase during axial compression load with the knee flexed, such as what may occur during ambulation. Moreover, static MRI protocols with the knee in full extension may underestimate the degree of extrusion in PRTs of the medial meniscus. On the other hand, these data suggest that axial compression load in flexed knee angles results in less extrusion in PRTs of the lateral meniscus.

While this model successfully demonstrates that PRTs increase meniscus extrusion, particularly in the medial meniscus, under axial compression load, we also examined the behavior of the meniscus when subjected to dynamically flexing the knee. Similar to axial compression load testing results, dynamic flexion of the knee resulted in differing behavior of the medial compared to the lateral meniscus with an incompetent posterior root. We found that when the posterior root was intact and competent, there was a direct linear relationship between knee flexion and meniscus extrusion for both the medial and lateral meniscus. Detaching the posterior root resulted in a nonlinear relationship between knee flexion and meniscus extrusion for the medial meniscus. Although this relationship is complex, it is clear that across lower degrees of knee flexion, meniscus extrusion increases rapidly and plateaus at knee flexion angles greater than about 50 degrees. The lateral meniscus responded differently after posterior root detachment. Although there was more scatter and an increase in the average extrusion during knee range motion with an incompetent posterior root, a direct linear relationship was preserved. Furthermore, compared to the medial meniscus, the amount of extrusion was significantly less for the lateral meniscus at knee flexion angles greater than 10-degrees.

Together these findings suggest that PRTs compromise the functional integrity of both menisci and result in an increase in meniscus extrusion compared to a competent posterior root consistent with previous biomechanical studies. For example, Camarda et al. found that using three single loops enhanced the biomechanical behavior for suture repair [31]. Similarly, Daney et al. used a cadaveric model and found that using a centralization suture reduced extrusion and restored tibiofemoral contact mechanics [12]. However, we found that the behavior of meniscus extrusion differed between the medial and lateral meniscus. Indeed, these findings suggest that while reducing meniscus extrusion is a requirement of suture repair, it may not be sufficient to recapitulate native function.

Here, we identified that the essential differences after PRTs between the medial and lateral meniscus are the extrusion and behavior of extrusion during motion. This study is the first to directly compare the extrusion between the medial and lateral meniscus with both a competent and incompetent posterior root condition. Moreover, we demonstrate that a nonlinear relationship exists in response to PRTs in the medial meniscus, which may be critical in optimizing repair following PRTs. Finally, we also demonstrate that the extrusion of the lateral meniscus is variable and results in significantly less extrusion in response to PRTs. The clinical significance of these findings are that PRTs of the lateral meniscus may not result in the same biomechanical changes as is found after PRTs of the medial meniscus, assuming a stable ACL and meniscofemoral ligament. While cadaveric tissue may not replicate that of living tissue, our model had no occurrences of tissue failure.

Similarly, we recognize that this is a "time-zero" study and does not directly consider the effects of any biologic healing, concomitant surgical procedures (e.g., anterior cruciate ligament reconstruction), or lower extremity alignment variability. However, at present, the characterization of meniscus extrusion *in vivo* is limited. Despite these limitations, this study further demonstrates the need to improve the fixation of posterior meniscus roots, particularly following medial root tears.

## Conclusions

Minimal extrusion occurs with a competent posterior meniscus root for both the medial and lateral menisci. The functional behavior associated with a detached posterior meniscus root are different between the medial and lateral meniscus. While strictly limiting the amount of meniscus extrusion after PRTs is necessary, it may not be sufficient to recapitulate native meniscus function.

## Supporting information

**S1 Table. Descriptive data of tested specimens.** yr = age of the specimen in years. (DOCX)

## Acknowledgments

The authors thank Dr. Ellen Leiferman, D.V.M. for her help in preparing the specimens for testing.

## Author Contributions

**Conceptualization:** Brian E. Walczak, Lisa Sienkiewicz, Heather Hartwig Stokes, Ron McCabe, Geoffrey S. Baer.

**Data curation:** Brian E. Walczak, Kyle Miller, Michael A. Behun, Lisa Sienkiewicz, Heather Hartwig Stokes, Ron McCabe.

**Formal analysis:** Brian E. Walczak, Kyle Miller, Michael A. Behun, Lisa Sienkiewicz, Heather Hartwig Stokes, Ron McCabe.

**Funding acquisition:** Brian E. Walczak.

**Investigation:** Brian E. Walczak, Kyle Miller, Michael A. Behun, Lisa Sienkiewicz, Heather Hartwig Stokes, Ron McCabe.

**Methodology:** Brian E. Walczak, Ron McCabe, Geoffrey S. Baer.

**Software:** Ron McCabe.

**Validation:** Heather Hartwig Stokes, Ron McCabe.

**Writing – original draft:** Brian E. Walczak, Kyle Miller, Heather Hartwig Stokes, Ron McCabe.

**Writing – review & editing:** Brian E. Walczak, Heather Hartwig Stokes.

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
