## [Decision Letter · Decision Letter 0]

3 Sep 2021

PONE-D-21-16714Quantifying the Functional Consequences Between the Medial and Lateral Meniscus After Posterior Meniscus Root TearsPLOS ONE

Dear Dr. Walczak,

Thank you for submitting your manuscript to PLOS ONE. After careful consideration, we feel that it has merit but does not fully meet PLOS ONE’s publication criteria as it currently stands. Therefore, we invite you to submit a revised version of the manuscript that addresses the points raised during the review process.

We look forward to receiving your revised manuscript.

Kind regards,

Osama Farouk

Academic Editor

PLOS ONE

Dr. Walczak was supported by NIH UW T32 AG000213-26, NIH UW TL1 TR000429, NIH UW UL1 TR000427, and UW-Madison Department of Orthopedic Surgery’s Freedom of Movement Award. No conflicts of interest, financial or otherwise, are declared by the authors.”

“B.E.W. was supported by NIH UW T32 AG000213-26, NIH UW TL1 TR000429, NIH UW UL1 TR000427, and UW-Madison Department of Orthopedic Surgery’s Freedom of Movement Award.”

“I have read the journal's policy and the authors of this manuscript have the following competing interests: B.E.W. is a consultant for AlloSource and G.S.B. is a consultant for Conmed.”

6. Thank you for stating the following in the Competing Interests section:

“I have read the journal's policy and the authors of this manuscript have the following competing interests: B.E.W. is a consultant for AlloSource and G.S.B. is a consultant for Conmed.”

We note that one or more of the authors are employed by a commercial company: Conmed

“The funder provided support in the form of salaries for authors B.E.W but did not have any additional role in the study design, data collection and analysis, decision to publish, or preparation of the manuscript. The specific roles of these authors are articulated in the ‘author contributions’ section.”

Within your Competing Interests Statement, please confirm that this commercial affiliation does not alter your adherence to all PLOS ONE policies on sharing data and materials by including the following statement: ""This does not alter our adherence to  PLOS ONE policies on sharing data and materials.” (as detailed online in our guide for authors http://journals.plos.org/plosone/s/competing-interests) . If this adherence statement is not accurate and  there are restrictions on sharing of data and/or materials, please state these. Please note that we cannot proceed with consideration of your article until this information has been declared.

7. We note that you have stated that you will provide repository information for your data at acceptance. Should your manuscript be accepted for publication, we will hold it until you provide the relevant accession numbers or DOIs necessary to access your data. If you wish to make changes to your Data Availability statement, please describe these changes in your cover letter and we will update your Data Availability statement to reflect the information you provide.

8. Your ethics statement should only appear in the Methods section of your manuscript. If your ethics statement is written in any section besides the Methods, please delete it from any other section.

Additional Editor Comments (if provided):

Reviewers' comments:

Reviewer's Responses to Questions

**Comments to the Author**

1. Is the manuscript technically sound, and do the data support the conclusions?

Reviewer #1: Yes

Reviewer #2: Partly

2. Has the statistical analysis been performed appropriately and rigorously? 

Reviewer #1: Yes

Reviewer #2: I Don't Know

3. Have the authors made all data underlying the findings in their manuscript fully available?

Reviewer #1: Yes

Reviewer #2: No

4. Is the manuscript presented in an intelligible fashion and written in standard English?

Reviewer #1: Yes

Reviewer #2: Yes

5. Review Comments to the Author

Reviewer #1: Comment on the manuscript Number PONE-D-21-16714

Quantifying the Functional Consequences Between the Medial and Lateral Meniscus After Posterior Meniscus Root Tears

• The idea of the research is interesting for the readers of the journal. The authors used a cadaveric model to compare the effect of the meniscal root detachment of the medial and lateral meniscus on the meniscal extrusion. Using a cadaveric model, the test was carried out on both medial and lateral menisci during constant axial compression load, simulated axial load in various knee flexion angles (0-90°) as well as simulated non-weightbearing ROM (0-100°). The test was repeated while the meniscal root intact and after detaching the posterior meniscal roots.

• The study methodology is very clear. The results are reported in sufficient details. The statistical analysis used in the study is sound.

• The article is written in a clear and simple English language. I did not found any spelling or grammatical mistakes.

• With exception of figure 1, the included Diagrams and table are informative.

• The abstract is short and comprehensive and summarizes the important points of the article.

• The authors reported the weak points and the limitation of the study.

• Only few improvement suggestions:

­ line 195: I think the authors meant lateral meniscus not medial.

­ Figure 1 is of low quality and very low resolution/Pixel. For example you cannot read the flexion angle written in part C. I recommend to replace with pictures of higher resolution.

• Decision:

According to the guidelines of the critical appraisal, I recommend to accept this Article for publication after considering the previously mentions improvement recommendations.

Reviewer #2: Quantifying the Functional Consequences Between the Medial and Lateral Meniscus

After Posterior Meniscus Root Tears

Aim of the study:

The objective of this study was to understand essential differences in the response to posterior root tears (PRTs ) between the medial and lateral meniscus.

the authors conducted a cadaveric study to quantify the subluxation of both medial and lateral meniscus both with intact and with torn posterior root. This quantification was performed in 3 different settings; 1: axial compression load, 2: varying degrees of flexion (0-90°) with sinusoidal loading to mimic gait 3: dynamic flexion with no load (non weight bearing). The degree of subluxation was compared both between the intact and torn posterior horn within each meniscus and across both menisci.

This is a novel work with interesting findings. They provided a good introduction. The methodology was described in details. Statistical analysis was also described. Power calculation for sample size was conducted. The results are detailed and discussed adequately. Limitations of the study were also discussed.

points to be improved.

The title should be modified for example : Quantifying the differential Functional Consequences Between the Medial and Lateral Meniscus

After Posterior Meniscus Root Tears.

The abstract could be more simple and highlight the novelty of this work. It should be more attractive to let the reader continue into the article. The conclusion that functional consequences of extrusion are more significant

for the medial meniscus than that of the lateral meniscus however can not be simply made based on the degree of meniscus subluxation. Other factors that were not measured in this work such as changes in stress distribution across the joint surface with PRT could make such conclusion possible.

The software used for statistical analysis should be mentioned

In the results section, how was the overall subluxation in table 1 measured? Please provide a supplementary table with all measures for all specimens.

Line 176, 177 : The greatest extrusion was recorded at 30-degrees of knee flexion (4.000 ± 01.259mm), please add the statistical significance . also in the same paragraph ensure that the statistical significance of each difference (which can be luckily found in the figures) is always mentioned

Line 195: lateral meniscus instead of medial ?

In the conclusion, as mentioned above in the comment on the abstract, no functional consequences of extrusion were tested here, as for example the differential stress distribution across the joint in response to this measured meniscus subluxation. So the authors can not make this assumption. The direct fact that can be concluded here is obviously the significant subluxation of medial meniscus in the different given testing situations in comparison to the lateral meniscus.

6. PLOS authors have the option to publish the peer review history of their article (what does this mean?). If published, this will include your full peer review and any attached files.

Reviewer #1: **Yes: **Ayman F. AbdelKawi

Reviewer #2: **Yes: **Dr. Mohammad Masoud

---

## [Author Response · Author response to Decision Letter 0]

26 Sep 2021

Rebuttal

Manuscript ID # PONE-D-21-16714

Title: Quantifying the Differential Consequences Between the Medial and Lateral Meniscus After Posterior Meniscus Root Tears

We would like to thank the reviewers for their insightful comments and suggestions. We have taken the suggestions into consideration and revised the manuscript accordingly to address each of the reviewers’ questions and concerns. 

Editor Comments:

Author’s Response:

We appreciate the editor’s critical feedback of the formatting requirements and we have reviewed the requirements provided in the above link. To address the concern, we have revised the manuscript accordingly.

Author’s Action:

We have changed the title to sentence case, added the location of the laboratory where the experiments were performed, removed the symbols denoting ‘current address’ as it is the same as the affiliation denoted by the numerical superscripts, and we have removed the corresponding author’s name and replaced it with ‘* Corresponding author,’ as described in the formatting requirement template. 

Author’s Response:

We appreciate the editor’s critical feedback of the requirements for IRB or ethics approval. To address the concern, we have revised the manuscript by including the IRB approval and waiver of informed consent.

Author’s Action:

While the IRB approval organization and number was provided in the ethics statement in ‘editor manager,’ we have added the following to the beginning of the materials and methods section: “The University of Wisconsin-Madison Institutional Review Board approved the study under 2016-1316 and as a cadaveric study with anonymous data analysis written consent was not required.”

Author’s Response:

We appreciate the editor’s critical feedback of the requirements for the Funding Information and Financial Disclosure and we understand the importance of this information. To address this concern, we have revised the information accordingly and apologize for this confusion. 

Author’s Action:

We have removed the funding information within the text of the manuscript that was previously under ‘Acknowledgements’ to comply with the formatting described within the pdf provided titled, “MANUSCRIPT BODY FORMATTING GUIDELINES.” The funding information provided within the online submission system is appropriate funding for this study. The additional funding information can be found in the revised cover letter.

Dr. Walczak was supported by NIH UW T32 AG000213-26, NIH UW TL1 TR000429, NIH UW UL1 TR000427, and UW-Madison Department of Orthopedic Surgery’s Freedom of Movement Award. No conflicts of interest, financial or otherwise, are declared by the authors.”

“B.E.W. was supported by NIH UW T32 AG000213-26, NIH UW TL1 TR000429, NIH UW UL1 TR000427, and UW-Madison Department of Orthopedic Surgery’s Freedom of Movement Award.”

Author’s Response:

Again, we appreciate and thank the editorial review process for this critical feedback and again we understand this importance of this information, both in regards to accuracy and format. To address this concern, we have revised the information accordingly and apologize for this confusion. 

Author’s Action:

We have removed the funding information within the text of the manuscript that was previously under ‘Acknowledgements’ to comply with the formatting described within the pdf provided titled, “MANUSCRIPT BODY FORMATTING GUIDELINES.” We have placed in the ‘Acknowledgements’ section the person who assisted the work and she has approved her recognition in the acknowledgement section. Furthermore, the following is how we would like the Funding Statement to read, and is listed as ordered in the Funding Information section of the online submission site, “B.E.W. was supported by National Center for Advancing Translational Sciences under award numbers: UL1 TR000427, TL1 TR000429, and TL1 TR002375, National Institute on Aging under award number: T32 AG000213, and the Department of Orthopedic Surgery of the University of Wisconsin-Madison’s Freedom of Movement Award. The funders had no role in study design, data collection and analysis, decision to publish, or preparation of the manuscript.”

“I have read the journal's policy and the authors of this manuscript have the following competing interests: B.E.W. is a consultant for AlloSource and G.S.B. is a consultant for Conmed.”

Author’s Response:

Again, we appreciate and thank the editorial review process for this critical feedback regarding competing interests. We also want full transparency with any and all potential competing interests. 

Author’s Action:

We have added the confirmation that any competing interests declared do not alter adherence to the PLOS ONE policies on sharing data and materials. We have also added into our cover letter the updated competing interest section as described and includes the following additional statement which accurately the author’s relationships, "This does not alter our adherence to PLOS ONE policies on sharing data and materials.”

6. Thank you for stating the following in the Competing Interests section:

“I have read the journal's policy and the authors of this manuscript have the following competing interests: B.E.W. is a consultant for AlloSource and G.S.B. is a consultant for Conmed.”

We note that one or more of the authors are employed by a commercial company: Conmed

“The funder provided support in the form of salaries for authors B.E.W but did not have any additional role in the study design, data collection and analysis, decision to publish, or preparation of the manuscript. The specific roles of these authors are articulated in the ‘author contributions’ section.”

Within your Competing Interests Statement, please confirm that this commercial affiliation does not alter your adherence to all PLOS ONE policies on sharing data and materials by including the following statement: ""This does not alter our adherence to PLOS ONE policies on sharing data and materials.” (as detailed online in our guide for authors http://journals.plos.org/plosone/s/competing-interests) . If this adherence statement is not accurate and there are restrictions on sharing of data and/or materials, please state these. Please note that we cannot proceed with consideration of your article until this information has been declared.

Author’s Response:

Again, we appreciate and thank the editorial review process for this critical feedback regarding competing interests. We also want full transparency with any and all potential competing interests. 

Author’s Action:

We have added revisions to the cover letter as follows: “I have read the journal's policy and the authors of this manuscript have the following competing interests: B.E.W. is a consultant for AlloSource and G.S.B. is a consultant for Conmed. This does not alter our adherence to PLOS ONE policies on sharing data and materials. The commercial affiliates had no role in study design, data collection and analysis, decision to publish, or preparation of the manuscript. The authors did not receive payment or renumeration in any form for any efforts related to this work.”

7. We note that you have stated that you will provide repository information for your data at acceptance. Should your manuscript be accepted for publication, we will hold it until you provide the relevant accession numbers or DOIs necessary to access your data. If you wish to make changes to your Data Availability statement, please describe these changes in your cover letter and we will update your Data Availability statement to reflect the information you provide.

Author’s Response:

We thank the editor’s response for repository information regarding the data. We have addressed the concerns and provided the relevant DOIs.

Author’s Action:

We have submitted the data to DRYAD and the data for this manuscript can be accessed by linking the following DOI: https://doi.org/10.5061/dryad.1ns1rn8v9.

8. Your ethics statement should only appear in the Methods section of your manuscript. If your ethics statement is written in any section besides the Methods, please delete it from any other section.

Author’s Response:

We understand and have revised the manuscript accordingly.

Author’s Action:

We added the ethical approval of this study in the first sentence in the methods section and removed from any other section.

Reviewers' comments:

Reviewer's Responses to Questions

Comments to the Author

1. Is the manuscript technically sound, and do the data support the conclusions?

Reviewer #1: Yes

Author’s Response:

We thank the reviewer for the positive comment.

Reviewer #2: Partly

Author’s Response:

We thank the reviewer for the essential feedback and address this concern below.

2. Has the statistical analysis been performed appropriately and rigorously?

Reviewer #1: Yes

Author’s Response:

We thank the reviewer for the positive comment.

Reviewer #2: I Don't Know

Author’s Response:

We thank the reviewer for their review. 

Author’s Action:

To make complete assurances regarding the analyses, we have published the data from this study and it can be downloaded from the link provided above https://doi.org/10.5061/dryad.1ns1rn8v9.

3. Have the authors made all data underlying the findings in their manuscript fully available?

Reviewer #1: Yes

Author’s Response:

We thank the reviewer for the positive comment.

Reviewer #2: No

Author’s Response:

We thank the reviewer for the feedback and we have now provided all data to DRYAD and is accessible on DOI: https://doi.org/10.5061/dryad.1ns1rn8v9.

4. Is the manuscript presented in an intelligible fashion and written in standard English?

Reviewer #1: Yes

Author’s Response:

We thank the reviewer for the positive comment.

Reviewer #2: Yes

Author’s Response:

We thank the reviewer for the positive comment.

5. Review Comments to the Author

Reviewer #1: Comment on the manuscript Number PONE-D-21-16714

Quantifying the Functional Consequences Between the Medial and Lateral Meniscus After Posterior Meniscus Root Tears

• The idea of the research is interesting for the readers of the journal. The authors used a cadaveric model to compare the effect of the meniscal root detachment of the medial and lateral meniscus on the meniscal extrusion. Using a cadaveric model, the test was carried out on both medial and lateral menisci during constant axial compression load, simulated axial load in various knee flexion angles (0-90°) as well as simulated non-weightbearing ROM (0-100°). The test was repeated while the meniscal root intact and after detaching the posterior meniscal roots.

Author’s Response:

We thank the reviewer for the positive comment.

• The study methodology is very clear. The results are reported in sufficient details. The statistical analysis used in the study is sound.

Author’s Response:

We thank the reviewer for the positive comment.

• The article is written in a clear and simple English language. I did not found any spelling or grammatical mistakes.

Author’s Response:

We thank the reviewer for the positive comment.

• With exception of figure 1, the included Diagrams and table are informative.

Author’s Response:

We thank the reviewer for the positive comment.

• The abstract is short and comprehensive and summarizes the important points of the article.

Author’s Response:

We thank the reviewer for the positive comment.

• The authors reported the weak points and the limitation of the study.

Author’s Response:

We thank the reviewer for the positive comment.

• Only few improvement suggestions:

­ line 195: I think the authors meant lateral meniscus not medial.

Author’s Response:

We thank the reviewer for their critical review of this. The typo is now corrected.

­ Figure 1 is of low quality and very low resolution/Pixel. For example you cannot read the flexion angle written in part C. I recommend to replace with pictures of higher resolution.

Author’s Response:

We thank the reviewer for their feedback and understand the concern and agree that the photographs can be improved.

Author’s Action:

Figure 1 has been revised with photographs of higher resolution for clarity. Additionally, we have added a high-resolution photo of the electronic goniometer capturing real-time angular measurements associated with meniscus extrusion. The figure legend has also been improved with more detail to illustrate the novel methods of this research.

• Decision:

According to the guidelines of the critical appraisal, I recommend to accept this Article for publication after considering the previously mentions improvement recommendations.

Author’s Response:

We thank the reviewer for their positive review.

Reviewer #2: Quantifying the Functional Consequences Between the Medial and Lateral Meniscus

After Posterior Meniscus Root Tears

Aim of the study:

The objective of this study was to understand essential differences in the response to posterior root tears (PRTs ) between the medial and lateral meniscus.

the authors conducted a cadaveric study to quantify the subluxation of both medial and lateral meniscus both with intact and with torn posterior root. This quantification was performed in 3 different settings; 1: axial compression load, 2: varying degrees of flexion (0-90°) with sinusoidal loading to mimic gait 3: dynamic flexion with no load (non weight bearing). The degree of subluxation was compared both between the intact and torn posterior horn within each meniscus and across both menisci.

This is a novel work with interesting findings. They provided a good introduction. The methodology was described in details. Statistical analysis was also described. Power calculation for sample size was conducted. The results are detailed and discussed adequately. Limitations of the study were also discussed.

points to be improved.

The title should be modified for example : Quantifying the differential Functional Consequences Between the Medial and Lateral Meniscus

After Posterior Meniscus Root Tears.

Author’s Response:

We welcome the reviewer’s critique and after considering the comments agree that the title should be modified to both more precisely characterize the purpose of this research and the findings. 

Author’s Action:

To address this concern, we have revised the title to more precisely characterize the purpose of the research and subsequent findings. The title has been modified to, “Quantifying the differential functional behavior between the medial and lateral meniscus after posterior meniscus root tears.”

The abstract could be more simple and highlight the novelty of this work. It should be more attractive to let the reader continue into the article. The conclusion that functional consequences of extrusion are more significant

for the medial meniscus than that of the lateral meniscus however can not be simply made based on the degree of meniscus subluxation. Other factors that were not measured in this work such as changes in stress distribution across the joint surface with PRT could make such conclusion possible.

Author’s Response:

We thank the reviewer for their comments and agree that the abstract could better highlight the novelty of the work and be more attractive to reader to continue into the paper. We have revised the abstract in response to address this concern.

Author’s Action:

We have simplified the abstract to highlighting the novelty of the research and encouraging the reader to find details within the paper. Additionally, we have clarified the conclusion stating that there is a differential functional behavior in extrusion between the medial and lateral meniscus and this must be accounted for in posterior root repair.

The software used for statistical analysis should be mentioned

Author’s Response:

We agree that the software used for statistical analysis should be mentioned and we thank the reviewer for the suggestion.

Author’s Action:

We have added the software used for statistical analysis at the end of the ‘Statistical analysis’ subsection of the ‘Materials and methods’ section.

In the results section, how was the overall subluxation in table 1 measured? Please provide a supplementary table with all measures for all specimens.

Author’s Response:

We thank the reviewer for pointing out an area of needed clarification. We have revised the manuscript to provide full access to the data and the first paragraph in the ‘Results’ section was expanded to clarify how the overall subluxation was calculated. 

Author’s Action:

To address this comment, we have provided the data using DRYAD accessible through https://datadryad.org/stash/share/XR2diRRfUYUd5cDw_d79mYO_i3iaJu0S5EW-FgcuMw8 and the manuscript was revised to include the doi number linked to the DRYAD repository where the full data is deposited and accessible upon publication of the research article in PLOS ONE at the end of the first paragraph in the ‘Results’ section. Table 1’s footnote was expanded to clarify the measurements of the Summary data.

Line 176, 177 : The greatest extrusion was recorded at 30-degrees of knee flexion (4.000 ± 01.259mm), please add the statistical significance . also in the same paragraph ensure that the statistical significance of each difference (which can be luckily found in the figures) is always mentioned

Author’s Response:

We agree with the reviewer that the narrative could be improved for clarity. We have revised the paragraph for the results regarding meniscus extrusion and knee flexion.

Author’s Action:

We have revised the figure 3 results narrative adding the statistical significance next to the point estimates and variation to augment the data provided in Figure 3.

Line 195: lateral meniscus instead of medial ?

Author’s Response:

We thank the reviewer for their critical review of this. The typo is now corrected.

In the conclusion, as mentioned above in the comment on the abstract, no functional consequences of extrusion were tested here, as for example the differential stress distribution across the joint in response to this measured meniscus subluxation. So the authors can not make this assumption. The direct fact that can be concluded here is obviously the significant subluxation of medial meniscus in the different given testing situations in comparison to the lateral meniscus.

Author’s Response:

We thank the reviewer for bringing up this point. We agree that in general the stress distribution across the joint is a useful piece of data. However, there are technical challenges in adding a sensor to measure stress distribution using this model, and we found that the sensor actually inhibited the dynamic motion of the meniscus in response to axial compression and range of motion, which was the focus of this work. Moreover, it is known that stress distribution and pressure is altered after detachment of the posterior root as many previous studies have demonstrated. For example, a seminal article showed that peak contact pressure after posterior root tear was similar to a total meniscectomy (Allaire et al., The Journal of Bone and Joint Surgery, 2008). A more recent study used a finite element analysis and demonstrated increased contact pressure and contact stress after posterior root avulsion (Wang et al., Journal of Orthopaedic Surgery and Research, 2021). 

Author’s Action:

The discussion section was revised by adding a discussion point of what is known regarding the stress distribution and pressure changes associated with posterior root avulsions. We have also simplified the conclusion to highlight that focus of the work regarding the influence of knee motion and axial compression on the differential meniscus extrusion between the medial and lateral meniscus.

6. PLOS authors have the option to publish the peer review history of their article (what does this mean?). If published, this will include your full peer review and any attached files.

Do you want your identity to be public for this peer review? For information about this choice, including consent withdrawal, please see our Privacy Policy.

Reviewer #1: Yes: Ayman F. AbdelKawi

Reviewer #2: Yes: Dr. Mohammad Masoud

---

## [Decision Letter · Decision Letter 1]

25 Oct 2021

Quantifying the differential functional behavior between the medial and lateral meniscus after posterior meniscus root tears

PONE-D-21-16714R1

Dear Dr. Walczak,

We’re pleased to inform you that your manuscript has been judged scientifically suitable for publication and will be formally accepted for publication once it meets all outstanding technical requirements.

Kind regards,

Osama Farouk

Academic Editor

PLOS ONE

Additional Editor Comments (optional):

Reviewers' comments:

Reviewer's Responses to Questions

**Comments to the Author**

1. If the authors have adequately addressed your comments raised in a previous round of review and you feel that this manuscript is now acceptable for publication, you may indicate that here to bypass the “Comments to the Author” section, enter your conflict of interest statement in the “Confidential to Editor” section, and submit your "Accept" recommendation.

Reviewer #1: All comments have been addressed

Reviewer #2: All comments have been addressed

2. Is the manuscript technically sound, and do the data support the conclusions?

Reviewer #1: Yes

Reviewer #2: Yes

3. Has the statistical analysis been performed appropriately and rigorously? 

Reviewer #1: Yes

Reviewer #2: Yes

4. Have the authors made all data underlying the findings in their manuscript fully available?

Reviewer #1: Yes

Reviewer #2: Yes

5. Is the manuscript presented in an intelligible fashion and written in standard English?

Reviewer #1: Yes

Reviewer #2: Yes

6. Review Comments to the Author

Reviewer #1: (No Response)

Reviewer #2: The authors have made all the needed and recommended changes to the article. I have no further comments.

7. PLOS authors have the option to publish the peer review history of their article (what does this mean?). If published, this will include your full peer review and any attached files.

Reviewer #1: **Yes: **Ayman F. AbdelKawi

Reviewer #2: **Yes: **Dr. Mohammad Masoud

---

## [Editor Report · Acceptance letter]

29 Oct 2021

PONE-D-21-16714R1 

Quantifying the differential functional behavior between the medial and lateral meniscus after posterior meniscus root tears 

Dear Dr. Walczak:

I'm pleased to inform you that your manuscript has been deemed suitable for publication in PLOS ONE. Congratulations! Your manuscript is now with our production department. 

Kind regards, 

on behalf of

Dr. Osama Farouk 

Academic Editor

PLOS ONE